# Development of a Novel Endometrial Signature Based on Endometrial microRNA for Determining the Optimal Timing for Embryo Transfer

**DOI:** 10.3390/biomedicines12030700

**Published:** 2024-03-21

**Authors:** Ching-Hung Chen, Farn Lu, Wen-Jui Yang, Wei-Ming Chen, Pok Eric Yang, Shih-Ting Kang, Tiffany Wang, Po-Chang Chang, Chi-Ting Feng, Jung-Hsuan Yang, Chen-Yu Liu, Chi-An Hsieh, Lily Hui-Ching Wang, Jack Yu-Jen Huang

**Affiliations:** 1Institute of Molecular and Cellular Biology, National Tsing Hua University, Hsinchu 300, Taiwan; chchen@taiwanivfgroup.com (C.-H.C.); lilywang@life.nthu.edu.tw (L.H.-C.W.); 2Department of Obstetrics and Gynecology, Ton Yen General Hospital, Hsinchu 302, Taiwan; farn.lu@gmail.com (F.L.); molinda11@yahoo.com.tw (W.-J.Y.); 3Taiwan IVF Group Center for Reproductive Medicine & Infertility, Hsinchu 302, Taiwanivfstork8@gmail.com (C.-A.H.); 4Inti Labs, Hsinchu 302, Taiwan; wmchen@intilabs.com (W.-M.C.); eric@intilabs.com (P.E.Y.); tiffany@intilabs.com (T.W.); sandyyang@intilabs.com (J.-H.Y.); 5Pharus Taiwan, Inc., Hsinchu 302, Taiwan; d94b41007@ntu.edu.tw; 6Quark Biosciences Taiwan, Inc., Hsinchu 302, Taiwan; 7Department of Medical Science, National Tsing Hua University, Hsinchu 300, Taiwan; 8School of Medicine, National Tsing Hua University, Hsinchu 300, Taiwan; 9Division of Reproductive Endocrinology and Infertility, Stanford University, Stanford, CA 94305, USA

**Keywords:** non-coding RNA, microRNAs, endometrial receptivity, infertility, multi-gene analysis

## Abstract

Though tremendous advances have been made in the field of in vitro fertilization (IVF), a portion of patients are still affected by embryo implantation failure issues. One of the most significant factors contributing to implantation failure is a uterine condition called displaced window of implantation (WOI), which refers to an unsynchronized endometrium and embryo transfer time for IVF patients. Previous studies have shown that microRNAs (miRNAs) can be important biomarkers in the reproductive process. In this study, we aim to develop a miRNA-based classifier to identify the WOI for optimal time for embryo transfer. A reproductive-related PanelChip^®^ was used to obtain the miRNA expression profiles from the 200 patients who underwent IVF treatment. In total, 143 out of the 167 miRNAs with amplification signals across 90% of the expression profiles were utilized to build a miRNA-based classifier. The microRNA-based classifier identified the optimal timing for embryo transfer with an accuracy of 93.9%, a sensitivity of 85.3%, and a specificity of 92.4% in the training set, and an accuracy of 88.5% in the testing set, showing high promise in accurately identifying the WOI for the optimal timing for embryo transfer.

## 1. Introduction

Globally, 8~12% of reproductive-aged couples suffer from infertility [1]. To solve this, affected couples have turned to assisted reproductive technologies, such as in vitro fertilization (IVF), to address this issue. However, there are still factors that contribute to the low success rates of IVF treatment. Successful embryo implantation during IVF treatment requires using good quality embryo(s) and implanting the embryo at the optimal time for the endometrium [2]. The optimal time for the endometrium refers to the receptive state of the endometrium [3]. An endometrium is receptive only for a relatively short period each month, referred to more commonly as the window of implantation (WOI). This usually occurs around days 19–20 of the menstrual cycle and lasts 4–5 days [4].

Various studies have shown that a displaced window of implantation (WOI) is one of the reasons for implantation failure [2,5,6]. By identifying the WOI, embryo transfer can be performed at the endometrium’s optimal time, increasing patients’ chances of a successful IVF cycle [2,5,6].

Conventionally, histological dating methods and ultrasound assessments are used to assess the receptivity status of the endometrium [7,8,9,10,11]. However, these methods cannot clearly distinguish the receptive and non-receptive phases of the endometrium [12,13,14,15,16]. Consequently, the gene expression profiling of endometrial tissue to determine the different stages of endometrial receptivity has been developed using high-throughput technologies [17,18,19]. However, there are still several issues to consider when using high-throughput technologies such as next-generation sequencing (NGS) and microarray technologies. For example, NGS-based RNA sequencing requires a larger portion of endometrial tissue, higher RNA quality and integrity, and has a longer turnaround time [20,21,22]. Moreover, reproducibility and bias are also major concerns for NGS-based RNA sequencing [23,24].

MicroRNAs (miRNAs) are involved in the regulatory mechanisms that indicate the WOI [25,26]. For example, the miR-200 family regulates epithelial cell proliferation and differentiation during endometrial cyclic changes [27,28,29]. High levels of miR-145 have been associated with the suppression of embryo attachment by regulating type-1 insulin-like growth factor receptor (IGF1R) [30]. Moreover, miRNA profiling has shown different expression profiles between fertile and infertile sample donors [31,32,33]. Our previous study demonstrated that combining three miRNA signatures, hsa-miR-20b-5p, hsa-miR-155-5p, and hsa-miR-718, can differentiate patients with repeated implantation failure from a control group, indicating miRNAs’ capabilities in identifying displaced WOI [34]. 

This study aims to develop a miRNA-based classifier to identify the optimal time for embryo transfer or the WOI by utilizing endometrial biopsy samples from patients who underwent IVF treatment, followed by personalized embryo transfer with the guidance of endometrial receptivity analysis testing.

## 2. Materials and Methods

### 2.1. Endometrial Samples

In a hormone replacement therapy (HRT) cycle, the endometrium reaches the desired thickness of >7 mm and it is confirmed that there is no leading follicle via ultrasound. Following this, progesterone supplementation is commenced. The dosing and routes of administration of progesterone are 800 mg orally or vaginally and an 80 mg vaginal suppository daily. The endometrial tissue samples were collected from the uterine cavity using a Pipelle catheter (Gynétics, Lommel, Belgium) after 120 ± 3 h of progesterone administration. The endometrial tissue sample was divided into two sections and stored in two cryotubes containing 1.5 mL of RNAlater™ Stabilization Solution (Thermo Fisher Scientific, Waltham, MA, USA). After storage at 4 °C for at least 8 h, one of the cryotubes was sent to an external laboratory to undergo endometrial receptivity testing. The other cryotube containing the spare endometrial tissue samples was stored at the study site at −20 °C as a backup during sample shipment in case of sample degradation.

Infertile patients undergoing assisted reproductive technology (ART) treatment and endometrial biopsy to assess the window of implantation following personalized embryo transfers with stocks of remaining samples were invited to participate in this study.

### 2.2. Ethical Approval

The utilization of the remaining endometrial tissue samples for developing a microRNA classifier was approved by the Joint Institutional Review Board (JIRB No. 19-S-012-1). Written informed consent was obtained from all participants.

### 2.3. Study Design

For this study, the remaining endometrial biopsy tissue samples from 200 patients who underwent the HRT cycle for IVF treatment and personalized embryo transfer following the results of endometrial receptivity analysis were collected and analyzed using the reproductive-related PanelChip^®^ (Quark Biosciences Taiwan, Inc., Hsinchu, Taiwan) to obtain their miRNA expression profiles. The period of sample collection was between September 2019 and September 2022. In total, 150 out of the 200 patients experienced successful implantation and live birth following the recommendations of the endometrial receptivity analysis. The successful implantation samples were subsequently chosen to build a miRNA-based classifier (Figure 1, Table 1) from the miRNA expression profiles outputted from PanelChip^®^. First, these 150 samples were further randomly separated into two datasets, a training set (75% of the dataset, N = 115) and a testing set (25% of the dataset, N = 35). The patients in the training set were split into three groups based on the results of the endometrial receptivity analysis (Figure 1): 108 ± 5 h, 120 ± 5 h, and 144 ± 5 h. The miRNA-based classifier was built using the training set samples (Figure 2), and the performance of the miRNA-based classifier was evaluated based on the testing set samples. The remaining 50 patients who had failed embryo implantation were processed for further statistical analysis to compare whether or not the miRNA-based classifier had a different classification compared to the original endometrial receptivity analysis.

### 2.4. RNA Extraction and miRNA Enrichment

Total RNAs were isolated from endometrial tissue using the miRNeasy Micro Kit (QIAGEN, Hilden, Germany) as described in the manufacturer’s instruction manual. Briefly, 5 milligrams (mg) of endometrial tissue was disrupted and homogenized in liquid nitrogen with a mortar and pestle. Next, 700 µL of QIAzol Lysis Reagent was added to the homogenized tissue, which was incubated at room temperature for 5 min to promote the dissociation of nucleoprotein complexes. After the incubation, the sample was transferred to a tube, followed by adding 140 μL of chloroform. The tube was shaken vigorously for 15 s and incubated at room temperature for 3 min. Next, the sample was centrifuged at 12,000× *g* for 15 min at 4 °C, with the resulting upper aqueous phase transferred to a new tube. One volume of 70% ethanol was added to the tube and vortexed thoroughly. The sample was added to an RNeasy MinElute spin column and centrifuged at 8000× *g* for 15 s at room temperature. Flow-through was pipetted to a 2 mL tube and vortexed thoroughly after adding 0.65 volumes of 100% ethanol. The mix was transferred to an RNeasy MinElute spin column and centrifuged at 8000× *g* for 15 s at room temperature. The following column washing process with centrifugation was performed: (1) 700 μL of Buffer RWT; (2) 500 μL of Buffer RPE; and (3) 500 μL of 80% ethanol. Finally, the column was centrifuged for 5 min at 8000× *g*, and miRNAs were eluted with 14~20 μL of nuclease-free for 1 min at 8000× *g*. The miRNA-enriched fraction was stored at −80 °C for future analysis.

### 2.5. cDNA Synthesis

Two or more nanograms of the miRNA-enriched fraction from endometrial tissue was used to synthesize cDNA in a 20 µL reverse transcription reaction. According to the manufacturer’s instructions, reverse transcription was performed using the microRNA Universal RT Kit (Quark Biosciences Taiwan, Inc., Hsinchu, Taiwan). Briefly, the Poly-A tail was added to the miRNA using Poly-A polymerase, which was performed according to the following program: 42 °C for 60 min and 95 °C for 5 min, and 4 °C after that. The cDNA was stored at −20 °C for future analysis.

### 2.6. miRNA Expression Analysis

A previous study [34] introduced the reproductive-related PanelChip^®^ of miRNA candidates involved in reproductive diseases such as previous implantation failure (pIF). The reproductive-related PanelChip^®^ consisted of two PanelChips containing 167 miRNA biomarkers, 3 endogenous controls (RNU6B, RNU43, and 18s rRNA), and 3 exogenous spike-in controls used to monitor extraction, cDNA synthesis, and qPCR efficiency (Quark Biosciences Taiwan, Inc., Taiwan). In total, 143 of the 167 miRNAs have been shown to be related to 257 genes involved in the reproductive process, embryo development, organ development, and cytokinesis (Appendix A). The reproductive-related PanelChip^®^ was preloaded with miRNA-specific primer sets [35]. The cDNA was analyzed with the reproductive-related PanelChip^®^. cDNA (equivalent to 0.1 ng of the miRNA-enriched fraction) was added to the mixture containing 30 μL of 2X SYBR Master Mix (Quark Biosciences Taiwan, Inc., Taiwan), and nuclease-free water was added to the mixture to obtain a final volume of 60 μL. The mixture was mixed by hand thoroughly and briefly spun down to collect the liquid at the bottom. Then, 60 μL of the mixture was dispensed along the edge of the chip, and the mixture was then applied across the entire surface of the reproductive-related PanelChip^®^ via a scraping motion with a glass slide. The PanelChip was then submerged in a tray containing Channeling Solution (Quark Biosciences Taiwan, Inc., Taiwan), with reaction wells facing the bottom of the tray. Each tray was then placed into a Q Station™ for signal detection [36] according to the following program: 95 °C for 36 s and 60 °C for 72 s for 40 cycles. 

### 2.7. Data Processing and Analysis

A total of 200 patients were recruited in this retrospective study (Figure 1 and Table 1). Data from the reproductive-related PanelChip^®^ were obtained by calculating the average per sample and preprocessing. A total of 526 expression profiles were normalized using the quantile normalization method [37]. After averaging the replicates, 115 of the 200 profiles were chosen for the training set to build a miRNA-based classifier (Figure 1 and Figure 2, and Table 1). For data preprocessing, 24 out of the 167 miRNAs without amplification signals across 10% of the profiles were removed; the remaining 143 miRNAs were used as features in the classifier; the missing miRNA values for individual profiles were replaced by the maximum ΔCq of all profiles [38]. For subsequent data preprocessing, each feature in the expression profiling was scaled by zero-mean and unit-variance. The PCA-based feature transformation was applied to extract meaningful features [39]. Based on the embryo transfer times, miRNA expression profiles were categorized into three classes: 108 ± 5 h, 120 ± 5 h, and 144 ± 5 h. The R package glmnet in version 4.1, containing Elastic Net regularized generalized linear models [40], was used to build the miRNA-based classifier. Ten-fold cross-validation was used to finetune the parameters to improve the accuracy of the miRNA-based classifier. Thirty-five patients were chosen as the testing set (Figure 1 and Table 1) to evaluate the performance of the resulting miRNA-based classifier. The remaining 50 patients whose samples were not used due to failed implantation results were also processed for further analysis.

To identify differentially expressed miRNAs for 120 ± 5 h vs. 108 ± 5 h and 120 ± 5 h vs. 144 ± 5 h in the training set, two criteria, log2(fold-change) ≥ ±0.585 and *p*-value ≤ 0.05, were applied. The fold-changes were measured at 120 ± 5 h/108 ± 5 h and 120 ± 5 h/144 ± 5 h, with the *p*-value calculated using the unpaired Student *t*-test (or with an equal variance when two distributions were normal). To reduce the false discovery rate, miRNAs with no expression detected in more than 5% of samples of each group (108 ± 5 h, 120 ± 5 h, and 144 ± 5 h) were excluded [33]. Out of the 143 miRNAs used for analysis, 21 were differentially expressed. Based on these differentially expressed miRNAs, a PCA-based cluster was performed using R package stats version 3.4.2 and Factoextra version v1.0.6.

For in silico network analysis, the miRNA and mRNA regulatory network was confirmed using miRTarBase v8 [41] and manually curated for reproductive-related genes [18,42]. The biological processes were analyzed by AmiGO [43]. The miRNA–mRNA networks were visualized using Cytoscape version 3.8.0 [44].

### 2.8. Model Building

In the modeling process (Figure 2), one or more of the following steps were taken to build and validate the algorithm: data normalization, data scaling, data transformation, prediction modeling, and cross-validation, where X from Equation (1) is a vector of the set of microRNA’s Cq expression profiling; X from Equation (2) is a vector of the set of normalized values from Equation (1); X from Equation (3) is a vector of the set of scaled values from Equation (2); and X from Equation (4) is a vector of the set of PCA transformed values from Equation (3) and builds a regularized regression model using the Elastic Net method. The model is able to predict the miRNA score.

Suppose there is a dataset M×N of microRNA’s Cq expression profiling. After data normalization Equation (1), data scaling Equation (2), and data transformation Equation (3), the dataset has n observations with p predictors. Let $y={(y1|\ldots \left|yn\right)}^{T}$ be the Response (set 120±5 h=0, 144±5 h=1, 108±5 h=−1) and X=(x1|…|xp) be the model matrix, where $xj={(x1|\ldots \left|xnj\right)}^{T}$, j=1,…, p, are the predictors.

We define the score as $y=f\left(X\in eq\left(C\right)\right)=X\beta +\varepsilon$, where β is the minimizer of equation defined by $\beta =argmin\left({\left|y-X\beta \right|}^{2}+{\alpha |\beta |}^{2}+\left(1-\alpha )\right|\beta_{1}|\right)$ and ε is an error term that stands for any influence being exerted on the microRNA variable, such as changes in environmental factors. Here, the function ${\alpha |\beta |}^{2}+\left(1-\alpha )\right|\beta_{1}|$ is the so-called elastic net penalty, which is a convex combination of the lasso and ridge penalty. β and ε are generated depending on what the dataset and model parameters are.

Finally, we set the endometrial state (ES) subject to
$$\mathrm{ES}=\left\{\begin{array}{c} 144\pm 5\;h,\;score>1\\ \;108\pm 5\;h,\;score<-1\\ 120\pm 5\;h,\;otherwise \end{array}\right.$$

In this study, the R package glmnet version 4.1, containing Elastic Net regularized generalized linear models, was used to build the miRNA-based classifier. Ten-fold cross-validation was used to fine-tune the parameters to improve the accuracy of the miRNA-based classifier.

## 3. Results

### 3.1. Evaluation of miRNA-Based Classifier’s Performance

In total, 143 out of the 167 miRNAs with amplification signals across 90% of the expression profiles were analyzed, resulting in 21 differentially expressed miRNAs. The 21 differentially expressed miRNAs were then used for PCA-based clustering based on three groups. The 120 ± 5 h vs. 108 ± 5 h and 120 ± 5 h vs. 144 ± 5 h groups’ miRNA signatures indicated that miRNAs have the potential for differentiating between the three different time points (Appendix A). Therefore, we decided to utilize this set of 143 miRNAs to build a miRNA-based classifier based on the resulting miRNA expression profiles from the reproductive-related PanelChip^®^ analysis (Materials and Methods and Figure 2). 

In this study, we selected samples from patients who underwent successful implantation and grouped the samples into three different time ranges based on embryo transfer time (108 ± 5 h, 120 ± 5 h, and 144 ± 5 h). The 150 selected samples were divided into a training set (75% of the dataset, N = 115) and a testing set (25% of the dataset, N = 35). With regard to the characteristics of the 150 patients, the “age” and “no. of pIF” exhibited significant differences, indicating that one of the groups could stochastically dominate other groups (Table 1 and Appendix A). Based on the 10-fold cross-validation, the training set was used to train the miRNA-based classifier and finetune its parameters to improve the performance of the miRNA-based classifier (Figure 2). The miRNA-based classifier prediction achieved 93.9% accuracy, a sensitivity of 85.3% and a specificity of 92.4% in the training set (Table 2). Finally, we used a testing set to assess the performance of the miRNA-based classifier. With the testing set, the miRNA-based classifier matched the proper classification in 88.5% of samples (Table 2), indicating that the miRNA-based classifier can predict WOI with an accuracy of 88.5%. 

### 3.2. Analysis of Inconsistent Results

miRNAs are differentially expressed in the endometrium of IVF patients with and without recurrent implantation failure [31,32,33]. One of the factors that cause implantation failure is performing embryo transfer at a time when the endometrium is not mature enough to provide the embryo(s) with sufficient nutrition to implant and grow [2,5,6]. In this retrospective study, 50 patients had undergone personalized embryo transfer (pET) based on endometrial receptivity analysis but the implantation failed following the embryo transfer time suggested by the analysis. Overall, 19 of these 50 unsuccessful embryo transfer patients had utilized preimplantation genetic testing (PGT-A), while 11 of the 19 patients had suffered from pIF. When the samples were analyzed using the miRNA-based classifier, we obtained a concordance rate of 87.5% (7/8, No. of pIF = 0, see Figure 3). The concordance rate for the unsuccessful embryo transfer samples dropped to 63.1% (12/19, No. of pIF ≥ 0, see Figure 3), 45.4% (5/11, No. of pIF ≥ 1, see Figure 3), and 37.5% (3/8, No. of pIF ≥ 2, see Figure 3), indicating that the miRNA-based classifier had an increasingly higher chance of having a different interpretation than other endometrial receptivity analysis for pIF patients as the number of implantation failures increased.

## 4. Discussion

When the endometrium is not synchronized with the embryo transfer timing, there is a higher chance of implantation failure [2,5,6]. Various publications have shown that miRNAs can indicate a displaced window of implantation [31,32,33].

Key developments have been made in the past few years in the field of IVF treatment for personalizing embryo transfer by identifying the window of implantation (WOI) [45]. Still, both traditional methodologies and current molecular testing methods have certain drawbacks in terms of consistency and sensitivity [46]. While traditional methodologies can be critical indicators for comprehending physiological conditions, they cannot pinpoint the WOI, and interpretations may vary depending on the person performing the analysis [47]. On the other hand, with current molecular testing methods based on messenger RNA biomarkers run on NGS or qPCR platforms, questions have been raised regarding consistency due to the instability of messenger RNA biomarkers, leading to concerns regarding sensitivity and accuracy [48,49]. Notably, miRNAs can be recovered from different tissue sample resources, which supports their high degree of stability, even in degraded samples [50]. Accordingly, miRNAs are stable and robust biomarkers in clinics. 

In this study, we built and validated a comprehensive miRNA-based model for identifying endometrial receptivity that shows potential in predicting the window of implantation by taking validated and novel miRNAs related to reproductive health (Appendix A) and showing that the resulting expression profiles are able to differentiate three different time points (108 ± 5 h, 120 ± 5 h, and 144 ± 5 h) during the secretory period of a hormone replacement therapy cycle (Table 2). 

miRNA biomarkers in endometrial receptivity testing offer advantages such as greater stability, requiring less genetic material, and exhibiting a lower failure rate. Overall, miRNA-based tests are more efficient than mRNA-based alternatives.

The discrepancies between the miRNA-based classifier and the recommended embryo transfer time based on endometrial receptivity analysis indicate that further prospective studies are needed to confirm whether the miRNA-based classifier can successfully predict the WOI. In contrast, other solutions for identifying pET are unable to. The inconsistent results may be due to the samples being old archival samples, some of which were preserved for more than one year, causing slight differences in the resulting analysis of the expression profiles [51]. Another possibility we explore further is the length of the window of implantation for infertility patients, which also might cause the differing results of other endometrial receptivity testing methods compared to the miRNA-based model. Previous studies utilizing mRNA-based screening methods for pET have observed similar trends, where patients that have undergone both PGT-A and endometrial receptivity testing still could not achieve a successful pregnancy [52], indicating other potential factors affecting embryo plantation, such as immunological factors, thyroid functions, and chronic endometritis [53].

The testing set shows that the miRNA-based model can be a predictor for endometrial receptivity when using traditional clinical measurement methods as a baseline. The main limitation of this study is a limited sample size, with sample collection from a single study site, which may result in biased results. One limitation of this study is the smaller number of 144 ± 5 h samples (n = 2) in the validation dataset. Consequently, the results may reflect less pronounced analytical performance. For instance, the sensitivity of the 144 ± 5 h group stands at 50.0% (Table 2). Although the analytical performance for the 144 ± 5 h group is 83.3%, a more extensive validation with a larger sample size is imperative to thoroughly evaluate the model’s performance. Despite the constrained sample size, the experimental model employs miRNA expression patterns in tissue samples as a diagnostic marker to assess the receptivity status in the endometrium. Additionally, a prospective study is needed to fully understand the potential of the miRNA-based panel chosen and confirm whether the miRNA-based panel can surpass other technologies’ performance in predicting optimal embryo transfer time and WOI. In the current study, samples are collected from a single site without accounting for other physiological conditions that may affect reproductive success, such as endometriosis, endometritis, and polycystic ovary syndrome (PCOS). These physiological conditions have also been found to affect miRNA expression levels; therefore, patients with a relevant clinical history may not be accounted for in the current miRNA-based model due to altered miRNA expression levels. Combined with the small sample size, these factors may result in bias and may not accurately reflect the effects of the miRNA-based model in the real world. Hence, other than a prospective study, samples from multiple clinical sites with patient criteria accounting for additional physiological conditions and patient history should be carried out to confirm the findings in this paper. 

From further network analyses (Appendix A), we can also see that the selected miRNA biomarkers regulate mRNA biomarkers involved in tissue and organ development, embryo and maternal crosstalk, preeclampsia, and preterm delivery (Appendix A). This network analysis provides insights for potential future studies that may indicate not only the miRNAs chosen to reflect the WOI, but also if the successful implantation can develop into a healthy pregnancy and live birth by considering downstream regulatory mechanisms.

Additionally, the 21 differentially expressed miRNAs in the miRNA-based classifier could be seen to have unique expression during the three different pET time points (Appendix A), showing the phenomenon of a time course of the miRNA expression changes that could be occurring during other time points of the endometrium in the secretory phase.

## 5. Conclusions

In this study, we provide a new method for identifying the WOI by utilizing miRNA biomarkers instead of the more commonly seen messenger RNA profiling method via sequencing. Based on the miRNA expression profiles obtained using the reproductive-related PanelChip^®^, a miRNA-based classifier was built and achieved 93.9% accuracy in the training set and 88.5% accuracy in the testing set. From these results, we see that the miRNA-based classifier could potentially be used as an alternative screening tool for finding the WOI in patients undergoing IVF treatment.

While a larger sample size in a prospective study is needed to confirm the predictive accuracy of the model, we see its potential in separating and identifying the different stages that the endometrium cycles through within the secretory period. This novel testing method shows advantages such as a smaller tissue sample required, lowering the discomfort for patients during the sample collection process. Additionally, targeting specific miRNA biomarkers for analysis instead of performing whole-genome sequencing analysis can also reduce testing costs, lowering the financial burden for patients in the future. Combined with the natural advantages that miRNAs provide, such as higher stability and reproducibility, this method provides a more stable, less invasive, and lower-cost solution for patients to surpass other methodologies in identifying the optimal time for embryo transfer during IVF treatment.

## Figures and Tables

**Figure 1 biomedicines-12-00700-f001:**
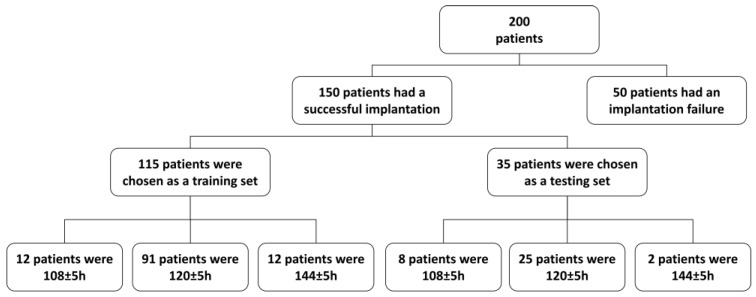
The selection criteria for the training and validation sets.

**Figure 2 biomedicines-12-00700-f002:**
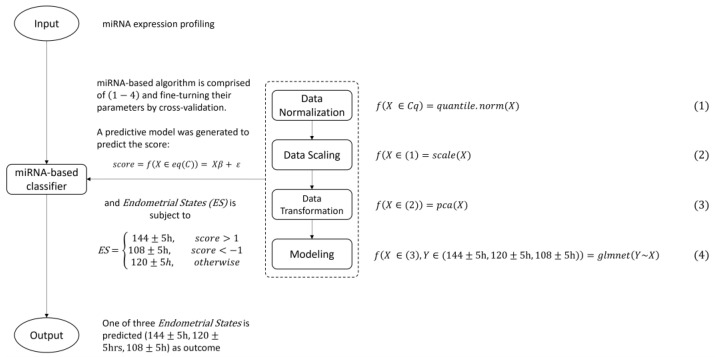
The framework of the miRNA-based classifier. The miRNA-based algorithm comprises data normalization, data scaling, data transformation, and modeling. The algorithm uses a miRNA-based classifier to find the WOI and finetunes its parameters by means of cross-validation to improve the accuracy.

**Figure 3 biomedicines-12-00700-f003:**
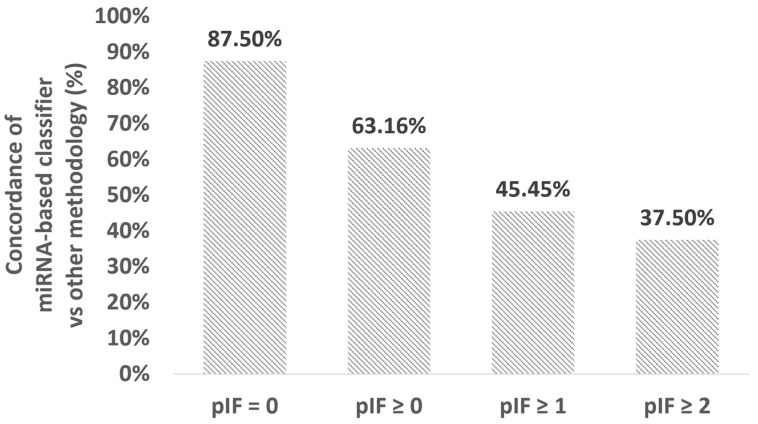
Concordance in WOI classification was obtained from the miRNA-based classifier and other methodologies. pIF: previous implantation failure.

**Table 1 biomedicines-12-00700-t001:** Characteristics of patients at the time of biopsy.

Characteristics	Training Set (N = 115)			Testing Set (N = 35)		
	108 ± 5 h	120 ± 5 h	144 ± 5 h	108 ± 5 h	120 ± 5 h	144 ± 5 h
	(N = 12)	(N = 91)	(N = 12)	(N = 8)	(N = 25)	(N = 2)
Age (years)						
Mean (SD)	36.1 (±2.8)	36.5 (±4.2)	39.8 (±4.6)	35 (±3)	38.6 (±4.4)	42.5 (±6.1)
Range	32–40	23–47	31–46	31–39	30–47	32–50
No. of previous implantation failure (#1)						
Mean (SD)	1.3 (±1.5)	0.3 (±0.8)	0.2 (±0.4)	1.3 (±2)	1 (±1.3)	1 (±0.9)
Range	0–5	0–4	0–1	0–5	0–5	0–3
Body mass index (BMI)						
Mean (SD)	20.7 (±1.4)	22.3 (±3.4)	22.9 (±3.2)	22.7 (±1.8)	22.8 (±2.9)	22.2 (±2.9)
Range	18.6–23.1	16.9–32	17.7–28.9	20.5–26.2	18–29.3	17.9–28.4
P4 Level (ng/mL) before exogenous progesterone administration						
Mean (SD)	0.3 (±0.3)	0.3 (±0.1)	0.3 (±0.1)	0.5 (±0.7)	0.3 (±0.2)	0.5 (±0)
Range	0.15–1.1	0.1–0.7	0.05–0.5	0.15–2.2	0.15–0.74	0.05–0.5
Endometrial thickness (mm)						
Mean (SD)	10 (±2.6)	10.7 (±2.4)	10.1 (±2.4)	10.1 (±2.3)	10.4 (±2.5)	9.6 (±1.9)
Range	6.05–15.1	5.8–20.9	7–15	7.7–14.2	7.2–19.2	7.9–14.9
A normal uterine cavity by office hysteroscopy	Yes	Yes	Yes	Yes	Yes	Yes

Training set: a group of patients was used to develop a miRNA-based classifier to determine the optimal timing of embryo transfer. Testing set: a group of patients was used to validate the miRNA-based classifier. P4: Progesterone. #1: the number will count as 0 if the patient doesn’t provide the record of previous implantation failure.

**Table 2 biomedicines-12-00700-t002:** Confusion matrix for the classification of the miRNA-based classifier. A confusion matrix is a table used to define a classification algorithm’s performance. Each row of the matrix represents the instances in an actual class, while each column represents the instances in a predicted class.

		**Actual Class (Training Set)**							
		**108 ± 5 h**	**120 ± 5 h**	**144 ± 5** **h**	**SEN**	**SPE**	**PPV**	**NPV**	**ACC**
**Predicted Class**	**108 ± 5 h**	9	1	0	75.00%	99.03%	90.00%	97.14%	-
	**120 ± 5 h**	3	89	2	97.80%	79.17%	94.68%	90.48%	-
	**144 ± 5 h**	0	1	10	83.33%	99.03%	90.91%	98.08%	-
					85.38%	92.41%	91.86%	95.23%	93.91%
		**Actual Class (Testing Set)**							
		**108 ± 5 h**	**120 ± 5 h**	**144 ± 5 h**	**SEN**	**SPE**	**PPV**	**NPV**	**ACC**
**Predicted Class**	**108 ± 5 h**	7	1	0	87.50%	96.30%	87.50%	96.30%	-
	**120 ± 5 h**	1	23	1	92.00%	80.00%	92.00%	80.00%	-
	**144 ± 5 h**	0	1	1	50.00%	96.97%	50.00%	96.97%	-
					76.50%	91.09%	76.50%	91.09%	88.57%
108 ± 5 h = the optimal time of 108 h (±5 h) for embryo transfer. 120 ± 5 h = the optimal time of 120 h (±5 h) for embryo transfer. 144 ± 5 h = the optimal time of 144 h (±5 h) for embryo transfer. Actual Class = the actual clinical status of the subjects. Predicted Class = the predicted clinical status of the subjects using the algorithm. TP = True Positives. FP = False Positives. FN = False Negatives. TN = True Negatives. Sensitivity (SEN) = TP / (TP + FN). Specificity (SPE) = TN / (TN + FP). Positive Predictive Value (PPV) = TP / (TP + FP). Negative Predictive Value (NPV) = TN / (TN + FN). Accuracy (ACC) = (TP + TN) / (TP + FP + TN + FN)									

## Data Availability

The dataset for constructing the predictive classifier can be downloaded from https://doi.org/10.5281/zenodo.10791280, accessed on 12 March 2024.

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
