# Peer review of "Development of a Novel Endometrial Signature Based on Endometrial microRNA for Determining the Optimal Timing for Embryo Transfer"

_biomedicines, 2024, doi:10.3390/biomedicines12030700_

Round 1

Reviewer 1 Report

Comments and Suggestions for Authors

The authors report on the development of a novel endometrial signature based on endometrial microRNA for determining the optimal timing for embryo transfer.

They found that the microRNA-based classifier identified the optimal timing for embryo transfer with an accuracy of 93.9%, a sensitivity of 85.3%, and a specificity of 92.4% in the training set, and an accuracy of 88.5% in the testing set, showing high promise in accurately identifying the optimal timing for embryo transfer.

I fully agree with the authors’ statemen that: “One of the significant factors contributing to implantation failure is a uterine condition called the displaced window of implantation (WOI), which results in an unsynchronized endometrium and embryo transfer time for IVF patients during treatment.”.

I consider that the present manuscript is a detailed study, based on complex and high scientific techniques and with a good statistical analysis.

Author Response

Reply: We thank the reviewer for this appreciation.

Reviewer 2 Report

Comments and Suggestions for Authors

The authors performed an analysis of endometrial microRNAs (miRNAs) in women undergoing assisted reproductive technology to establish a novel miRNA-based classifier for determining the optimal timing for embryo transfer. The infertile patients underwent endometrial receptivity analysis followed by in vitro fertilization (IVF) and the authors conducted the verification of the accuracy, sensitivity, and specificity of the miRNA-based classifier. They concluded that the miRNA-based classifier could be an useful screening tool to find the window of implantation (WOI) in patients undergoing IVF treatment.

Although the content of this paper is interesting and novel and the methodology and rationale of this study seems well written, I have some major concerns about this article.

1. The results of ‘training set’ is significantly important in this study. The authors should present those results more clearly and more concisely with some figures in the manuscript. 

2. Related to figure 3, it is difficult to compare the accuracy of endometrial receptivity analysis and their results in this paper. The result of endometrial receptivity analysis is fundamentally not accurate because it is also decided followed by classification only with the patterns of gene expression profiles in the endometrium, which may vary even in the same patient in the different cycle. If they discuss the accuracy, they must show the success rate of embryo implantation with the optimal WOI decided after their analysis. If they do not have such data, this figure must be deleted.

3. The authors should discuss the advantages of this method over messengerRNA-based classifiers and whether this method is clinically implementable in terms of the degree of invasiveness to the patient, the amount of sample required, and the cost.

4. The authors should show the period of sample collection in this study.

5. The authors should show the administration route of progestogens in the patient characteristics.

Author Response

We thank the reviewer for their helpful comments.

We are pleased to hear that the manuscript is found to be highly interesting and important in the subject matter. We believe that the issues that the reviewer has raised are generally straightforward to address satisfactorily, and to this effect, we have substantially revised the manuscript. Please find below a detailed point-by-point response to all comments (reviewers’ comments in black, our replies in blue). Since the reordering and restructuring of the manuscript was substantial, Line numbering refers to the revised manuscript, attached as a supplement to the Editor's Response.

  1. The results of ‘training set’ is significantly important in this study. The authors should present those results more clearly and more concisely with some figures in the manuscript. 

Reply: We thank you for this suggestion. We have added a section to the manuscript to provide a clearer understanding of how to use the training set, establish the model, and make other statements.

The updated version is as follows:

Line 218-241: 2.8 Model building

In the modeling process (Figure 2), one or more of the following steps are taken to build and validate the algorithm: data normalization, data scaling, data transformation, prediction modeling, and cross-validation, where X from Eq(A) is a vector of the set of microRNA’s Cq expression profiling; X from Eq(B) is a vector of the set of normalized values from Eq(A); X from Eq(C) is a vector of the set of scaled values from Eq(B); X from Eq(D) is a vector of the set of PCA transformed values from Eq(C) and builds a regularized regression model using Elastic Net method. The model is able to predict the MIRA score.

Suppose there is a data set of microRNA’s Cq expression profiling. After data normalization Eq(A), data scaling Eq(B), and data transformation Eq(C), the data set has  observations with  predictors. Let  be the Response () and  be the model matrix, where , , are the predictors.

We define score:, where  is the minimizer of equation defined by , and ,an error term, is stand for any influence being exerted on the microRNA variable, such as changes in environmental factor. Here the function  is so-called the elastic net penalty, which is a convex combination of the lasso and ridge penalty.  and  are generated depending on what dataset and model parameters are.

                        Finally, we set is subject to

In this study, the R package glmnet in version 4.1, elastic-net regularized generalized linear models, was used to build the miRNA-based classifier. 10-fold cross-validation was used to fine-tune the parameters to improve the accuracy of the miRNA-based classifier.

  1. Related to Figure 3, it is difficult to compare the accuracy of endometrial receptivity analysis and their results in this paper. The result of endometrial receptivity analysis is fundamentally not accurate because it is also decided followed by classification only with the patterns of gene expression profiles in the endometrium, which may vary even in the same patient in the different cycle. If they discuss the accuracy, they must show the success rate of embryo implantation with the optimal WOI decided after their analysis. If they do not have such data, this figure must be deleted.

Reply: We have amended the text and Figure 3 to clarify the concordance rate with other methodologies.

The updated version now reads as follows:

Line 290-291: Figure 3. Concordance in WOI classification was obtained from the miRNA-based classifier and other methodologies. pIF: previous implantation failure.

Line 282-288: When the samples were analyzed using the miRNA-based classifier, it resulted in a concordance rate of 87.5% (7/8, No. of previous implantation failure (pIF) = 0), indicating that previous implantation failure patients may be caused by other factors [53] rather than caused by a displaced window of implantation. The concordance rate for the unsuccessful embryo transfer samples dropped to 63.1% (12/19, No. of pIF >= 0), 45.4% (5/11, No. of pIF >= 1), and 37.5% (3/8, No. of pIF >= 2), indicating that previous implantation failure patients may not adjust to an optimal timing for embryo transfer. The results above indicate that miRNA-based classifier had an increasingly higher chance of having a different interpretation than other endometrial receptivity analysis for previous implantation failure patients as the number of implantation failures increased.

  1. The authors should discuss the advantages of this method over messengerRNA-based classifiers and whether this method is clinically implementable in terms of the degree of invasiveness to the patient, the amount of sample required, and the cost.

Reply: In this study, We used the remaining endometrial biopsy tissue samples from 200 patients who underwent the HRT cycle for IVF treatment and performed personalized embryo transfer following the results of endometrial receptivity analysis were collected and analyzed using the reproductive-related PanelChip® to obtain their miRNA expression profiles. For this reason, we don’t establish the cost and evaluate the degree of invasiveness to the patient.

The use of miRNA biomarkers in endometrial receptivity testing offers several advantages, including greater stability compared to mRNA using NGS or microarray, requiring less genetic material, and exhibiting a lower failure rate. There are some key points as below:

Reliability: miRNA is a stable testing material, allowing tests to be conducted with smaller tissue samples and a reduced failure rate.

Sensitivity: in this study, the method requires only 2mg of tissue from a biopsy, significantly less than mRNA-based tests, and offers a quicker turnaround time of 10 days.

Accuracy: miRNA profiles have an 88% success rate in predicting a patient's Window of Implantation (WOI), a crucial factor in successful IVF treatment, especially for those with implantation failure.

Overall, miRNA-based tests demonstrate superior success rates and efficiency in endometrial receptivity testing compared to mRNA-based alternatives. 

We addressed those advantages in the discussion. The updated version now reads as follows:

Line 314-316: miRNA biomarkers in endometrial receptivity testing offer advantages such as greater stability, requiring less genetic material, and exhibiting a lower failure rate. Overall, miRNA-based tests are more efficient than mRNA-based alternatives.

  1. The authors should show the period of sample collection in this study.

Reply: The sample collection time is from September 2019 to September 2022.

We have added a description on the materials and methods. The updated version now reads as follows:

Line 99: The period of sample collection was between September 2019 to September 2022.

  1. The authors should show the administration route of progestogens in the patient characteristics.

Reply: In our study, the HRT protocol is according to the genetic company of Igenomix. When the endometrium has reached the desired thickness of >7 mm and confirmed there is no leading follicle by ultrasound. Following the progesterone supplementation is commenced. The dosing and routes of administration of progesterone are 800 mg, oral or vaginal, and 80 mg vaginal suppository daily usage. An endometrial biopsy is performed on day P +5 (120±3 hours) and sent for an endometrial receptivity analysis. All patients are administered the same dosage of progesterone.

We have added a description on the material and methods. The updated version now reads as follows:

Line 72-77: In a hormone replacement therapy (HRT) cycle, the endometrium reached the desired thickness of >7 mm and confirmed no leading follicle by ultrasound. Following the progesterone supplementation is commenced. The dosing and routes of administration of progesterone are 800 mg, oral or vaginal, and 80 mg vaginal suppository daily use. The endometrial tissue samples were collected from the uterine cavity using a Pipelle catheter (Gynétics, Belgium) after 120±3 hours of progesterone administration.

Reviewer 3 Report

Comments and Suggestions for Authors

The manuscript biomedicines-2881006 aims to develop a miRNA-based classifier identifying the optimal timing for embryo transfer. A classifier was developed using a reproductive-related PanelChip, and the authors achieved to produce acceptable accuracy rate the testing set, suggesting potential for IVF patients.

Major comment 1.

Limitations of the study are not discussed.

Additionally, the authors highlight the potential advantages of this testing, however, it is important to consider the additional costs, time for procedure, and potential impact on patients associated with non-100% accurate test, etc.

Major comment 2.

In particular, the study population is insufficiently characterized, introducing significant risk of bias: (A) miRNA expression may be affected by infertility-related conditions such as uterine fibroids, adenomyosis, and endometriosis, PCOS, ... (B) The hormonal treatments used for assisted reproductive technology have not been accounted for. (C) Ethnic/genetic background differences have not been considered.

These significant biases need to be addressed in the present study. A multivariate analysis should be performed.

Major comment 3.

The current manuscript misses the authors’ vision. Please, put the study in perspectives: e.g. suggest an algorithm that ensures the rational and efficient implementation of this test, taking into account factors such as patient demographics, clinical history, and the specific reproductive context.

Major comment 4.

In Table2, sensitivity on testing set is reported 76%, lower than training set. Looking at subgroups, the sensitivity value is only 50% for group 144+-5h. This important result is eluded from the abstract, body of text and discussion. While reporting accuracy is informative, the sensitivity of the test is critical. Please state and discuss accordingly.

Minor comments

- The discussion should not repeat the results (delete l289-298). Clear all redundancy and emphasize the discussion on novelty, controversies, perspectives, ...

- Quality of figures should be enhanced (Fig2 namely).

- Doi: 10.5281/zenodo.10612264 : DOI NOT FOUND

Comments on the Quality of English Language

The style of English language needs revision.

All abbreviations must be defined (pIF).

Author Response

We thank the reviewer for their helpful comments.

We are pleased to hear that the manuscript is found to be highly interesting and important in the subject matter. We believe that the issues that the reviewer has raised are generally straightforward to address satisfactorily, and to this effect, we have substantially revised the manuscript. Please find below a detailed point-by-point response to all comments (reviewers’ comments in black, our replies in blue). Since the reordering and restructuring of the manuscript was substantial, Line numbering refers to the revised manuscript, attached as a supplement to the Editor's Response.

Major comment 1.

Limitations of the study are not discussed.

Additionally, the authors highlight the potential advantages of this testing, however, it is important to consider the additional costs, time for procedure, and potential impact on patients associated with non-100% accurate test, etc.

Reply:The limited sample size and collection from a single site may have resulted in biased results, which is the main limitation of this study. Further rigorous, well-designed studies are imperative to substantiate the miRNA-based classifier as proof-of-concept.

Major comment 2.

In particular, the study population is insufficiently characterized, introducing a significant risk of bias: (A) miRNA expression may be affected by infertility-related conditions such as uterine fibroids, adenomyosis, and endometriosis, PCOS, ... (B) The hormonal treatments used for assisted reproductive technology have not been accounted for. (C) Ethnic/genetic background differences have not been considered.

These significant biases need to be addressed in the present study. A multivariate analysis should be performed.

Reply: We didn't account for those variables in this study but might add them in the future. Your concern is duly acknowledged, and we appreciate the opportunity to address it, assuring the precision and accuracy of our study. We have addressed your concern of comment 1 and comment 2 by adding the sentence on discussion that describe the limitation of this study.

The updated version now reads as follows:

Line 332-334: The main limitation of this study is a limited sample size with sample collection from a single study site may result in biased results.

Major comment 3. 

The current manuscript misses the authors’ vision. Please, put the study in perspectives: e.g. suggest an algorithm that ensures the rational and efficient implementation of this test, taking into account factors such as patient demographics, clinical history, and the specific reproductive context.

Reply: The goal of this study is to create a classifier that uses miRNA to determine the best time for embryo transfer or the window of implantation (WOI). This will be achieved by analyzing endometrial biopsy samples from patients who have undergone IVF treatment. The patients will then receive personalized embryo transfers based on the results of endometrial receptivity analysis testing. The remaining endometrial biopsy tissue samples from 200 patients who underwent the HRT cycle for IVF treatment and performed personalized embryo transfer following the results of endometrial receptivity analysis were collected and analyzed using the reproductive-related PanelChip® to obtain their miRNA expression profiles. Out of the 200 patients, 150 had successful implantation and live birth following the endometrial receptivity analysis recommendation. The successful implantation samples were subsequently chosen for building a miRNA-based classifier. Regarding patient demographics, clinical history, and the specific reproductive context, we didn't account for those variables in this study due to the small sample size. Sufficient sample sizes are required for every factor to achieve statistical significance. We have identified some potential miRNA biomarkers for endometritis, endometriosis, PCOS, etc., but further studies are needed to confirm these findings.

Major comment 4.

In Table2, sensitivity on testing set is reported 76%, lower than training set. Looking at subgroups, the sensitivity value is only 50% for group 144+-5h. This important result is eluded from the abstract, body of text, and discussion. While report the test is critical. Please state and discuss accordingly.

Reply: We have added a description in the discussion. The updated version now reads as follows:

Line 334-341: One limitation of this study is the smaller number of 144+-5hr samples (n=2) in the validation dataset. Consequently, the results may reflect less pronounced analytical performance. For instance, the sensitivity of the 44+-5hr group stand at 50.0% (Table 2). Although the analytical performance for the 144+-hr group is 83.33.0%, a more extensive validation with a larger sample size is imperative to thoroughly evaluate the model's performance. Despite the constrained sample size, the experimental and model employing miRNA expression patterns in tissue samples as a diagnostic marker to assess receptivity status in the endometrium.

Minor comments

- The discussion should not repeat the results (delete l289-298). Clear all redundancy and emphasize the discussion on novelty, controversies, perspectives, ...

Reply: We have amended the manuscript and removed the paragraph from discussion in line 289-298. The content shows as follows:

As a retrospective study, we selected confirmed pregnancy samples with altered (144±5hrs and 108±5hrs) and non-altered (120±5hrs) embryo implantation times to build and train the miRNA-based model. 115 of the 200 clinical samples that were collected met the criteria to utilize as the training set for building the model. Through these samples, we achieved a cross-validation score of 93.9% (Table 2), showing that the selected miRNA biomarkers can be utilized as differentiators for the different stages of the endometrium during embryo implantation and can be successful predictors of the WOI. Then, for the testing set, we used 35 clinical samples with a successful pregnancy result following altered and non-altered embryo implantation times to validate the resulting miRNA-based model, resulting in an 88.5% accuracy rate (Table 2).

- Quality of figures should be enhanced (Fig2 namely).

Reply: The quality of Fig. 2 has been improved.

- Doi: 10.5281/zenodo.10612264 : DOI NOT FOUND

Reply: The dataset for constructing the predictive classifier can be downloaded from https://doi.org/10.17026/LS/YJ7PEQ, but it will be provided after the paper is accepted.

Comments on the Quality of English Language

The style of English language needs revision.

Reply: The English language style was revised by Tiffany Wang and Pok Eric Yang, who are Native Americans

All abbreviations must be defined (pIF).

Reply: pIF was defined as previous implantation failure. we have amended the revised manuscript to ensure consistent nomenclature.

The updated version now reads as follows:

Line 163-164: A previous study [34] introduced the reproductive-related PanelChip® of miRNA candidates involved in reproductive diseases such as previous implantation failure (pIF).

Line 257-259: With regards to the characteristics of 150 patients, the “age” and “no. of pIF” exhibited significant differences, indicating that one of the groups could stochastically dominate other groups (Table 1 and Supplemental Table 1).

Line 281-288: 11 of the 19 patients have suffered from pIF. When the samples were analyzed using the miRNA-based classifier, it resulted in a concordance rate of 87.5% (7/8, No. of pIF = 0, see Figure 3). The concordance rate for the unsuccessful embryo transfer samples dropped to 63.1% (12/19, No. of pIF >= 0, see Figure 3), 45.4% (5/11, No. of pIF >= 1, see Figure 3), and 37.5% (3/8, No. of pIF >= 2, see Figure 3), indicating that the miRNA-based classifier had an increasingly higher chance of having a different interpretation than other endometrial receptivity analysis for pIF patients as the number of implantation failures increased.

Round 2

Reviewer 2 Report

Comments and Suggestions for Authors

The authors revised the manuscript correctly according to the reviewer's suggestions.

Author Response

We appreciate the reviewer's careful reading of our manuscript and the insightful comments and suggestions.

Reviewer 3 Report

Comments and Suggestions for Authors

The study holds significance and warrants attention. The authors have addressed some of the critiques in the revised manuscript, but they have overlooked a substantial portion of the criticism and failed to incorporate several responses into the text. Please note that the reviewer does not seek personal responses but rather aims to provide constructive feedback to improve readability and impact.

Specifically, with respect to Major Comment 1 and MC2, the authors have not sufficiently revised the manuscript to address the significant study limitations and risk of bias, only superficially acknowledging the small sample size and single site (L332-334).

Furthermore, the authors missed the opportunity to revise the manuscript to provide context for the findings related to cost and perspectives. The possible additional cost and reduced expenses should at least be discussed (= contextualization and the authors' vision).

The content indicated at https://doi.org/10.17026/LS/YJ7PEQ is not accessible, thus impossible to evaluate. Please ensure the data is accessible for accurate review. Submission after acceptance does not comply with transparent data practices.

Comments on the Quality of English Language

L338: correct the percentage.

L339-341: sentence has no verb.

In the Author Contributions section, T.W. and P.E.W. should be acknowledged for proofreading the English language.

Author Response

We thank the reviewer for the helpful comments. we have substantially revised the manuscript. Please find below a detailed point-by-point response to all 
comments (reviewers’ comments in black, our replies in blue). 

The study holds significance and warrants attention. The authors have addressed some of the critiques in the revised manuscript, but they have overlooked a substantial portion of the criticism and failed to incorporate several responses into the text. Please note that the reviewer does not seek personal responses but rather aims to provide constructive feedback to improve readability and impact.

Specifically, with respect to Major Comment 1 and MC2, the authors have not sufficiently revised the manuscript to address the significant study limitations and risk of bias, only superficially acknowledging the small sample size and single site (L332-334).

Furthermore, the authors missed the opportunity to revise the manuscript to provide context for the findings related to cost and perspectives. The possible additional cost and reduced expenses should at least be discussed (= contextualization and the authors' vision).

Reply: Our study presents several limitations that need to be mentioned. It should be noted that our model could not identify all sources of variances, including other physiological conditions and ethnic/genetic background, which may potentially explain a small proportion of miRNA fluctuations between samples. The variances should be included in future studies. The sample collected from one site combined with the small sample size, these factors may result in bias and may not accurately reflect the effects of the miRNA-based model in the real world. To apply our proposed model to the entire female population, it may be necessary to investigate specific confounding factors in these cohorts. Your feedback and concerns are very valuable. We appreciate the opportunity to address them, assuring the precision and accuracy of our miRNA-based model.

The use of miRNA biomarkers in endometrial receptivity testing offers several advantages, including greater stability and lower costs compared to mRNA using NGS or microarray, requiring less genetic material, and exhibiting a lower failure rate. There are some key points below:

Reliability: miRNA is a stable testing material, allowing tests to be conducted with smaller tissue samples and a reduced failure rate.

Sensitivity: in this study, the method requires only 2mg of tissue from a biopsy, significantly less than mRNA-based tests, and offers a quicker turnaround time of 10 days.

Accuracy: miRNA profiles have an 88% success rate in predicting a patient's Window of Implantation (WOI), a crucial factor in successful IVF treatment, especially for those with implantation failure.

Overall, miRNA-based tests are more successful and efficient in testing endometrial receptivity than mRNA-based alternatives, and they are also less expensive.

We have addressed the description in the discussion and conclusion. Now the updated version reads as follows:

Line 342-352: In the current study, samples are collected from a single site without accounting for other physiological conditions that may affect reproductive success, such as endometriosis, endometritis, and polycystic ovary syndrome (PCOS). These physiological conditions have also been found to affect miRNA expression levels; therefore, patients with relevant clinical history may not be accounted for in the current miRNA-based model due to altered miRNA expression levels. Combined with the small sample size, these factors may result in bias and may not accurately reflect the effects of the miRNA-based model in the real world. Hence, other than a prospective study, samples from multiple clinical sites with patient criteria accounting for additional physiological conditions and patient history should be carried out to confirm the findings in this paper.

Line 375-383: This novel testing method shows advantages such as less tissue sample size required, lowering the discomfort for patients during the sample collection process. Additionally, targeting specific miRNA biomarkers to analyze instead of performing a whole genome sequencing analysis can also reduce testing costs, lowering the financial burden for patients in the future. Combined with the natural advantages that miRNAs provide, such as higher stability and reproducibility, this method provides a more stable, less invasive, and lower-cost solution for patients to surpass other methodologies in identifying the optimal time for embryo transfer during IVF treatment.

The content indicated at https://doi.org/10.17026/LS/YJ7PEQ is not accessible, thus impossible to evaluate. Please ensure the data is accessible for accurate review. Submission after acceptance does not comply with transparent data practices.

Reply: We have uploaded a dataset of the 526 expression profiles to https://doi.org/10.5281/zenodo.10791280 for supplementary data.

The updated version now reads as follows:

Line 185-186: A total of 526 expression profiles were normalized using the quantile normalization method [37].

Line 408-409: Data Availability Statement: The dataset for constructing the predictive classifier can be downloaded from https://doi.org/10.5281/zenodo.10791280.

Comments on the Quality of English Language

L338: correct the percentage

Reply: The percentage has been revised.

The updated version now reads as follows:

Line 335-337: Although the analytical performance for the 144±5hr group is 83.3%, a more extensive validation with a larger sample size is imperative to thoroughly evaluate the model's performance.

L337-339: sentence has no verb.

Reply: The sentence has been amended.

The updated version now reads as follows:

Line 339-341: Despite the constrained sample size, the experimental model employs miRNA expression patterns in tissue samples as a diagnostic marker to assess receptivity status in the endometrium.

In the Author Contributions section, T.W. and P.E.W. should be acknowledged for proofreading the English language.

Reply: Their contributions have been added to the manuscript.

The updated version now reads as follows:

Line 395-396: T.W., P.E.Y. —proofreading English language; 

Round 3

Reviewer 3 Report

Comments and Suggestions for Authors

All the comments have been addressed satisfactorily in the revised R2 version of the manuscript.